# Fluoride Affects Dopamine Metabolism and Causes Changes in the Expression of Dopamine Receptors (D1R and D2R) in Chosen Brain Structures of Morphine-Dependent Rats

**DOI:** 10.3390/ijms21072361

**Published:** 2020-03-29

**Authors:** Patrycja Kupnicka, Joanna Listos, Maciej Tarnowski, Agnieszka Kolasa-Wołosiuk, Agnieszka Wąsik, Agnieszka Łukomska, Katarzyna Barczak, Izabela Gutowska, Dariusz Chlubek, Irena Baranowska-Bosiacka

**Affiliations:** 1Department of Biochemistry and Medical Chemistry, Pomeranian Medical University in Szczecin, Powstańców Wlkp. 72, 70-111 Szczecin, Poland; patrycjakupnicka@o2.pl (P.K.); agnieszka_lukomska@wp.pl (A.Ł.); gutowska@pum.edu.pl (I.G.); dchlubek@pum.edu.pl (D.C.); 2Department of Pharmacology and Pharmacodynamics, Medical University of Lublin, Chodźki 4a, 20-093 Lublin, Poland; alistos@op.pl; 3Department of Physiology, Pomeranian Medical University, Powstańców Wlkp. 72, 70-111 Szczecin, Poland; maciejt@sci.pum.edu.pl; 4Department of Histology and Embryology, Pomeranian Medical University, Powstańców Wlkp. 72, 70-111 Szczecin, Poland; Agnieszka.Kolasa@pum.edu.pl; 5Maj Institute of Pharmacology PAS, Department of Neurochemistry, 12 Smętna Street, 31-343 Krakow, Poland; wasik@if-pan.krakow.pl; 6Department of Conservative Dentistry and Endodontics, Powstańców Wlkp. 72 St., Pomeranian Medical University in Szczecin, 70-111 Szczecin, Poland; kasiabarczak@vp.pl

**Keywords:** morphine, dependence, fluoride, dopamine, dopamine receptors

## Abstract

Disturbances caused by excess or shortages of certain elements can affect the cerebral reward system and may therefore modulate the processes associated with the development of dependence as was confirmed by behavioural studies on animals addicted to morphine. Earlier publications demonstrated and proved the neurodegenerative properties of both low and high doses of fluoride ions in animal experiments and in epidemiological and clinical studies. The aim of the experiments conducted in the course of the present study was to analyse the effect of pre- and postnatal exposure to 50 ppm F^−^ on the initiation/development of morphine dependence. For this purpose, the following were conducted: behavioural studies, the analysis of concentrations of dopamine and its metabolites, and the analyses of mRNA expression and dopamine receptor proteins D1 and D2 in the prefrontal cortex, striatum, hippocampus, and cerebellum of rats. In this study, it was observed for the first time that pre- and postnatal exposure to fluoride ions influenced the phenomenon of morphine dependence in a model expressing withdrawal symptoms. Behavioural, molecular, and neurochemical studies demonstrated that the degenerative changes caused by toxic activity of fluoride ions during the developmental period of the nervous system may impair the functioning of the dopaminergic pathway due to changes in dopamine concentration and in dopamine receptors. Moreover, the dopaminergic disturbances within the striatum and the cerebellum played a predominant role as both alterations of dopamine metabolism and profound alterations in striatal D1 and D2 receptors were discovered in these structures. The present study provides a new insight into a global problem showing direct associations between environmental factors and addictive disorders.

## 1. Introduction

Fluoride is a neurotoxic ion both in vitro and in vivo [1,2,3]. Chronic fluoride exposure has a negative impact on brain function, leading to neuronal apoptosis [4]. It leads to generation of free radicals, increases lipid peroxidation in the brain, and inhibits the production of antioxidant enzymes, therefore resulting in the formation of oxidative stress [5,6]. Exposure to this element is associated, among others, with a decrease in IQ levels, disturbances in the processes related to learning capabilities and memory, as well as a deterioration of psychomotor skills [2,7,8]. Accumulation of fluoride in the brain has been shown to cause neurodegenerative changes in the striatum, motor cortex, cerebellum, and amygdala of rats [9,10]. Studies on the influence of fluoride on the limbic system are limited. To date, it is recognised that long-term exposure to NaF results in an increase in dopamine levels in the striatum of rats as well as noradrenaline and serotonin levels in the hippocampus and neocortex [11,12].

The rewarding effect of various addictive substances is associated with dopaminergic stimulation of mesolimbic structures in brain. Dopamine acts on the D1-like family of dopamine receptors (D1 and D5) and the D2-like family (D2, D3 and D4). It is metabolized by the monoaminooxidase-B (MAO-B) enzyme or by catechol-*O*-methyl transferase (COMT) to 3,4-dihydroxyphenylacetic acid (DOPAC) or 3-methoxytyramine (3-MT), respectively, and then to the final metabolite—homovanillic acid (HVA). Exposure to abused drugs leads to an increased dopamine release in the nucleus accumbens (Di Chiara, 2000) which determines the feeling of pleasure.

Morphine is one of the strongest drugs of dependence that is used in clinical practice for acute and chronic pain alleviation. It acts primarily on μ opioid receptors. The stimulation of the μ receptor, localized at the neurons producing gamma-aminobutyric acid (GABAergic neurons) in the ventral tegmental area, inhibits GABA release. This leads to disinhibition of dopaminergic neurons, increases the release of dopamine, and causes euphoria, thus promoting the state of dependence [11]. On the other hand, the cessation of chronic administration of morphine reduces the dopamine level, which is manifested as the appearance of unpleasant signs characteristic of morphine withdrawal [12].

Drug dependence, including opioid dependence, is a global, clinical, social, and economic problem. The number of addicted patients is increasing steadily, while the possibilities of effective management are limited. Therefore, further studies are justified to elucidate the novel mechanisms underlying this issue which, in turn, may be helpful in searching for new therapeutic strategies. The main purpose of the present study is to assess whether prenatal and postnatal exposure to fluoride in rats influences morphine dependence. In the first step, behavioural experiments on rats were performed to show the influence of fluoride exposure on the intensity of morphine withdrawal. In the second stage, a series of neurochemical experiments was carried out, in which the levels of dopamine and its three metabolites (DOPAC, 3-MT, and HVA) in the hippocampus, the striatum, the prefrontal cortex, and the cerebellum were studied using high-performance liquid chromatography with the electrochemical detection (HPLC-ED) method. Next, mRNA and protein expressions of dopamine D1 and D2 receptors were estimated in the same brain structures using real-time quantitative reverse transcription PCR (qRT-PCR), the ELISA test, and immunohistochemical (IHC) reaction, respectively.

## 2. Results

### 2.1. Behavioural Studies

#### Effect of Prenatal and Postnatal Fluoride Exposure on the Intensity of Morphine Withdrawal Signs Observed as Jumps

The one-way ANOVA revealed significant changes in the behavioural study (F_3,49_ = 72.86, *p* < 0.0001). There were no jumps observed in saline and saline + naloxone (2 mg/kg, ip) rats. Similarly, no jumps were observed in animals pre- and postnatally exposed to fluoride and acute dose of naloxone. The exposure of rats to increasing doses of morphine and, on the last day of the study, to morphine with naloxone, produced a significant increase in the number of jumps (*p* < 0.001) in comparison with the saline + naloxone group. The prenatal and postnatal exposure of morphine-dependent rats to fluoride caused a significant increase (*p* < 0.001) in the number of jumps, both in comparison with fluoride-exposed rats and with morphine-treated rats without fluoride pre-exposure (Figure 1).

### 2.2. Ex Vivo Neurochemical Studies—the Analysis of the Concentration of Dopamine and Its Metabolites

Figure 2A shows the concentrations of dopamine as well as its metabolites in the analysed structures of the brain. In the prefrontal cortex, the statistical analysis showed that chronic exposure to morphine produced a significant decrease (*p* < 0.01) in the levels of dopamine and two dopamine metabolites–DOPAC and HVA (*p* < 0.01)—in comparison with saline-treated rats. However, it did not show any significant changes in dopamine and its metabolite concentrations in rats that were pre- and postnatally treated with fluoride. In morphine-withdrawal rats, which were previously treated with fluoride, significant reductions (*p* < 0.05) in the dopamine concentration and its two metabolites, DOPAC and HVA, were observed in comparison with morphine-or fluoride-treated rats. There were no significant changes in the 3-MT level in the prefrontal cortex of the studied rats.

In the striatum, in both groups, the following was observed: in animals exposed to fluoride and in morphine-withdrawal rats, a significant reduction (*p* < 0.05) in dopamine concentration, but not in that of dopamine metabolites, was observed in comparison with saline-treated rats. A significant reduction (*p* < 0.05) in the dopamine level in morphine-withdrawal rats, which were previously treated with fluoride, was observed in comparison with the rats treated with fluoride or morphine alone. The combination of fluoride pre-treatment with the exposure to morphine induced the increase (*p* < 0.05) in the level of DOPAC and HVA as compared to fluoride-treated rats. There were no significant changes in the striatal 3-MT level of the studied rats (Figure 2B).

In the hippocampus of animals exposed to morphine, there was a significant reduction (*p* < 0.05) in dopamine and DOPAC levels. The concentration of 3-MT was significantly increased (*p* < 0.05) in these animals. There were no significant changes in rats pre- and postnatally treated with fluoride as compared to the saline group. In morphine-withdrawal rats, which were previously treated with fluoride, significant reductions (*p* < 0.05) in dopamine, DOPAC, and HVA concentrations were demonstrated. In these animals, the level of 3-MT was significantly higher (*p* < 0.05) than that recorded in fluoride pre-treated rats and was significantly reduced (*p* < 0.05) in comparison with morphine-treated rats (Figure 2C).

In the cerebellum of fluoride pre-treated rats, the dopamine level was increased (*p* < 0.05) while the concentrations of dopamine metabolites (3-MT and HVA) were significantly (*p* < 0.05) reduced in comparison with saline-treated animals. In morphine-withdrawal rats, the level of dopamine was significantly elevated (*p* < 0.05), yet the concentrations of dopamine metabolites were diversified: DOPAC was significantly increased (*p* < 0.05), the HVA level was elevated (*p* < 0.05), and there was no significant alteration in the 3-MT level as compared to the saline-treated group. In rats which were treated with both fluoride and morphine, there was no change in the dopamine level in comparison with the animals treated with fluoride alone or morphine alone, while DOPAC and HVA levels were significantly reduced (*p* < 0.05) (Figure 2B).

### 2.3. The Analysis of Gene Expression of Dopamine Receptors in the Striatum, the Hippocampus, the Prefrontal Cortex, and the Cerebellum

The mRNA expression of the D1 dopamine receptor in the prefrontal cortex of the morphine group was more than 30 times lower than that in the control group (*p* < 0.05). In the M + F group, a pronounced down-regulation of the receptor was observed in comparison to the fluoride group (decrease of 97%) (*p* < 0.01). Administration of fluoride resulted in a twofold increase in the expression of D2 vs. C (*p* < 0.01). In the morphine group, a significant decrease in the expression of said receptor vs. C was observed (72-fold), similar to the M + F group vs. F (175-fold) (*p* < 0.01) (Figure 3).

The striatum of the fluoride group revealed an up-regulation of D1 and D2 receptor mRNA expression (5× and 1.5× resp., vs. C) (*p* < 0.05). Administration of morphine caused a down-regulation of the studied receptors vs. C (D1: 6× and D2: 1.75×) (*p* < 0.01). Pre-exposure to fluoride and subsequent morphine dependence resulted in a lowered expression of the D1 receptor vs. F among rats expressing withdrawal symptoms (*p* < 0.05), but did not affect the expression of the D2 receptor. The M + F group was also significantly different from the M group in terms of the expression of the analysed dopamine receptors (*p* < 0.01) (Figure 3).

No changes in the dopamine receptor mRNA expression in the hippocampus were observed between the fluoride group and the control group. As for M vs. C and MF vs. F, the D1R mRNA expression was markedly decreased (*p* < 0.05), similar to the expression of the D2 receptor (*p* < 0.05) (Figure 3).

Exposure to fluoride caused upregulation of mRNA expression of D1 and D2 in the cerebellum (40× and 35× vs. C, respectively) (*p* < 0.01 and *p* < 0.05, respectively). In the M group, the recorded expression level of the D1 and D2 receptors was, respectively, two times and ten times higher than the level recorded in the control group (*p* < 0.01 and *p* <0.05, respectively). The M + F group was characterised by a lower expression of the analysed receptors vs. F (*p* < 0.01 and *p* < 0.05, respectively) and, in the case of the D1 receptor, the expression was markedly higher than that observed in the M group (*p* < 0.01) (Figure 3).

### 2.4. The Analysis of the Protein Expression of Dopamine Receptors

The administration of fluoride alone did not affect the protein expression of the D1 receptor; however, the administration of morphine resulted in a decrease in the expression of said receptor in comparison to the control group (*p* < 0.05). The level of protein expression of D1 in the prefrontal cortex was 2 times lower in the M + F group vs. F (*p* < 0.05) and was 30% lower vs. M (*p* < 0.05). Exposure to fluoride resulted in an increase in D2 receptor expression by 40% (vs. C) (*p* < 0.05); the administration of morphine was associated with a decrease in the expression of said receptor vs. C (*p* < 0.05), similar to what was observed with respect to the MF vs. F groups (*p* < 0.05) (Figure 4).

The fluoride group showed a significantly higher D1 expression in the striatum as compared with the other groups (*p* < 0.05). In comparison to all the groups under study, the morphine group was characterised by the lowest D1R expression which was lower than that recorded in the control group, by 33% (*p* < 0.05).

D2 expression in the fluoride group was higher than that recorded in the C group, by 40% (*p* < 0.05). The M group was characterised by a significantly lower D2 expression in comparison to the C group (*p* < 0.05), whereas in the M + F group, the level of expression was comparable to that in the F group (Figure 4).

Exposure to fluoride did not result in significant changes in the protein expression of the D1 receptor (vs. C) in the structure of the hippocampus; however, it caused an increase in the expression of the D2 receptor, by 32% (*p* < 0.05). The protein expression of the D1 receptor was markedly lower in the M group as compared with the C group (*p* < 0.05). Exposure to F, when combined with morphine dependence, levelled the expression of D1 to that in the control group, whereas in the case of D2, the expression level was comparable to that of the M group (Figure 4).

Pre- and postnatal exposure to fluoride caused an increase in the expression of D1 vs. C, by 70% (*p* < 0.05), and of D2 by 110% (*p* < 0.05). Following the administration of naloxone, the morphine-dependent group (M) demonstrated lowered expression of D1R vs. C, by 23%. The M + F group was characterised by a lower level of D1 and D2 expression compared with the F group (*p* < 0.05) (Figure 4).

### 2.5. Immunohistochemical Analysis of Dopamine Receptors

#### 2.5.1. IHC for Dopamine Receptor D1

The IHC reaction showed that immunoexpression of dopamine receptor 1 in the prefrontal cortex (neocortex) was the highest in brains of fluoride-treated rats (Figure 5E, back arrows); in the control group (Figure 5A, back arrows), this expression was higher than in rats treated with morphine (Figure 5I, back arrows) and morphine + fluoride (Figure 5M, back arrows; Table 1).

Generally, the striatum portion of the studied rat brains showed a low expression of dopamine receptor 1 (Figure 5B,F,J,N), but the striatum from fluoride-treated rats (Figure 5F, back arrows) showed the highest DR1 intensity; in the control (Figure 5B, back arrows) and M + F (Figure 5N, back arrows) groups of rats, immunoactivity for DR1 was comparable, and the lowest/no reaction was observed in the striatum of morphine-treated rats (Figure 5J) (Table 1).

Figure 5C,G,K,O shows DR1-reactivity in the hippocampus proper of all 4 groups of studied animals (C, F, M, M + F), and this reaction seemed to be very similar in each group of rats. Mainly, the neurons of the granular cell layer (GCL) (black arrows) of dentate gyrus were DR1-positive; additionally, in the M + F group, a few neurons of the polymorphic cell layer (PCL) (green arrows) expressed DR1. In all groups, the neurons of the pyramidal cell layer (PyrCL) of the cornu ammonis were immunonegative (Table 1).

In the cerebellum of the studied animals (Figure 5D,H,L,P), DR1-immunoexpression manifested generally in the third layer of the cortex, in the granular cell layer (GCL) (black arrows). Also, some Purkinje cells (blue arrows) in the second layer of the cerebellar cortex of rats in the control (Figure 5D), morphine (Figure 5L), and morphine and fluoride (Figure 5P) groups were DR1-positive (Table 1).

#### 2.5.2. IHC for Dopamine Receptor D2

The IHC reaction showed that immunoexpression of dopamine receptor 2 in the prefrontal cortex (neocortex) seemed to be similar in all studied groups of animals (Figure 6A,F,I,M, black arrows) (Table 1).

The striatum of rats treated with fluoride (Figure 6F, black arrows) and morphine with fluoride (Figure 6N, black arrows) appeared to be most D2R-immunopositive; however, in the control group (Figure 6B, black arrows), the striatum had higher intensity of the IHC reaction compared with morphine-treated rats (Figure 6J, black arrows) (Table 1).

Figure 6C,G,K,O shows D2R-reactivity in the hippocampus proper of all 4 groups of studied animals (C, F, M, M + F), and this reaction seemed to be very similar in each group of rats. Moreover, the neurons of the pyramidal cell layer (PyrCL) of the cornu ammonis (yellow arrows) expressed higher DR2 levels than the neurons of the GCL of the dentate gyrus. In the hippocampus of the F (Figure 6G) and M + F (Figure 6O) groups of rats, polymorphic cells of the dentate gyrus (green arrows) were also D2R-positive (Table 1).

In the cerebellum cortex of the studied animals (Figure 6D,H,L,P), D2R-immunoexpression was manifested generally in the granular cell layer (GCL) (black arrows). A few Purkinje cells (blue arrows) in animals treated with fluoride (Figure 6H) and morphine (Figure 6L) were D2R-positive. Additionally, the granular cell layer of the cerebellum cortex of animals that were under intoxication (Figure 6H,L,P) exhibited histopathological changes (granular cells were swollen) (Table 1).

## 3. Discussion

In the present study, the effects of pre- and postnatal exposure to fluoride in a model of morphine dependence were studied in rats. A series of behavioural, neurochemical, and molecular experiments and microscopic observations was performed to assess the changes in the intensity of naloxone-induced morphine withdrawal signs in rats and to explore the changes within the dopaminergic system—the most important neurotransmitter in the CNS involved in addiction. To assess the effects of pre- and postnatal exposure to fluoride in rats, the animal model described by Kobayashi et al. (2011) and Morales-González et al. (2010) was chosen [13,14]. Accordingly, the animals were watered with fluoride solution at concentration of 50 ppm from conception to 60 PND. This concentration is considered as relatively low compared to the F^−^ levels observed in people living in areas rich in fluoride [15,16,17]. In the control rats, the fluoride concentration in drinking water did not exceed 1.5 mg/L, which was associated with obligatory standards in Poland [18].

It was observed in our previous studies, which were performed on a similar experimental model, that long-term fluoride exposure of rats did not produce any changes in the fluoride concentration in the serum, probably because of high urinary removal [19]; however, accumulation of this element was observed in some brain structures [20]. Thus, fluoride may act as a neurotoxic agent despite its low level in serum. It was documented that long-term exposure to fluoride causes tissue damage, free radical generation [1], and pro-inflammatory activity [20]. In people who were exposed to fluoride for a long period, the reduction of the intelligence quotient (IQ) and a decrease of cognitive and psychomotor functions [2,7,8] were observed. In another study, long-term fluoride-exposure increased levels of epinephrine, histamine, serotonin, and glutamate and decreased levels of norepinephrine, acetylcholine, and dopamine in a dose-dependent manner [21]. Thus, there are important reasons to perform further studies on the neurotoxic activity of fluoride. In light of current data on the increasing problem of global opioid dependence, it is important to verify whether fluoride exposure, as an environmental factor, may influence the dopaminergic system and the state of dependence.

Indeed, it was observed in a behavioural study that pre- and postnatal fluoride exposure significantly increased the intensity of morphine withdrawal signs, manifested as a higher number of jumps in comparison with the animals treated with morphine alone or fluoride alone. This confirms the role of the environmental factors in the development of the state of opioid dependence.

The neurochemical analysis showed that pre- and postnatal fluoride exposure produced the effects in the striatum and the cerebellum, but not in the hippocampus and the prefrontal cortex. In the striatum, a significant reduction in the dopamine concentration without the effects on dopamine metabolite levels was observed, while in the cerebellum, a significant increase in the dopamine level and a reduction of dopamine metabolites (3-MT and HVA) were measured. This showed that pre- and postnatal fluoride exposure disturbed the striatal dopamine concentration and inhibited dopamine degradation in the cerebellum. The literature data on the influence of fluoride exposure on dopamine concentration are not homogeneous. Chirumari and Reddy (2007) observed that intraperitoneal administration of fluoride for 14 days (1–10 mg/kg) caused an increase in the level of dopamine and its metabolite (HVA), as well as serotonin and its metabolite in the hippocampus and the neocortex [22]. On the other hand, rats receiving sodium fluoride (100 ppm) for 30 days, which were tested 15 days after fluoride discontinuation, showed an increase in dopamine concentration in the hippocampus and cortex [23]. Additionally, Tsunoda et al. (2005) did not show any significant changes in the concentration of dopamine and its metabolites in the cerebellum and the striatum after fluoride (1–125 ppm) exposure in drinking water [24].

In the case of morphine-dependent rats, during naloxone-induced morphine withdrawal, the inhibition of dopamine concentration in the striatum and the prefrontal cortex was demonstrated. In the hippocampus, the dopamine and DOPAC levels were reduced, while the 3-MT level was elevated. These divergent effects in the hippocampal dopamine metabolites resulted in the lack of changes in the concentration of the final metabolite—HVA. In the cerebellum, a marked increase in dopamine concentration was observed, but the changes in metabolite concentrations were strongly diversified, and some kind of artifact cannot be excluded in that part of the study. Generally, our striatal results confirmed the accepted theory that the appearance of unpleasant effects during morphine withdrawal is a consequence of a reduced dopamine level in the striatum [25]. The dopamine level in other structures in the brain during morphine withdrawal has not been precisely defined yet. For example, in our previous results [26] and those by other authors [27], no changes in the dopamine level in the hippocampus were observed. On the other hand, in the prefrontal cortex, an increase in the dopamine level was observed during morphine withdrawal [26,28]. Thus, the obtained hippocampal, prefrontal, and cerebellar results confirmed the involvement of dopaminergic mechanisms during morphine withdrawal, but the precise interactions need further studies.

One of the most important achievements in neurochemical studies is that pre- and postnatal fluoride exposure in morphine-withdrawal rats significantly disturbs dopamine metabolism, mainly in the striatum and cerebellum. In the striatum, the dopamine concentration was reduced and DOPAC and HVA levels were elevated, confirming the intensification of catabolic processes of dopamine, while in the cerebellum, the opposite effect was observed. The dopamine concentration was higher and its metabolites were reduced, which confirmed the reduction of dopamine degradation in these animals. Additionally, in the hippocampus and the prefrontal cortex of the studied animals, a reduction in dopamine levels, without an effect on dopamine metabolites, was observed in comparison with both fluoride- or morphine-treated rats. Thus, the obtained results demonstrated the existence of strong disturbances in dopamine release (all studied brain areas) and its degradation (striatum and cerebellum) in naloxone-induced morphine-withdrawal rats which were pre-treated with fluoride.

It was also demonstrated in neurochemical experiments that during naloxone-induced morphine withdrawal in fluoride-treated animals, the major enzyme involved in dopamine degradation is MAO, as significant alterations were measured in the levels of DOPAC and HVA. Statistical changes in the level of 3-MT, which is formed by degradation of dopamine by COMT, were observed only in the hippocampus of morphine-withdrawal rats. This effect was similar to our previous study [26] in which characteristic and significant increases in the hippocampal 3-MT level during naloxone-induced morphine withdrawal were also measured. There is a hypothesis that in some circumstances, 3-MT may act as a neuromodulator in CNS [29] and it may be an important factor of neuro-adaptive mechanisms [30]. It may be concluded from the present results that naloxone-evoked morphine discontinuation induces alterations in which 3-MT starts to act as a neuromodulator because of defence mechanisms arising in response to long-term fluoride exposure and morphine withdrawal.

As it was easy to predict, the alterations in the dopamine level evoked significant changes in dopamine D1 and/or D2 receptor expression in the studied brain structures. In fluoride-treated rats, despite a poor effect on dopamine concentration and its metabolism in a neurochemical study, a significant increase in mRNA and protein expressions in dopamine D1 and D2 receptors in most studied areas of the brain was observed. The Western blot analyses were consistent with the results obtained from immunohistochemical reactions. These changes might be a result of a long-term adaptation to fluoride exposure. On the other hand, after naloxone-evoked morphine discontinuation, a significant reduction in dopamine D1 and D2 receptors in most studied structures was observed in comparison with saline-treated rats. The pre- and postnatal fluoride exposure in morphine-withdrawal rats produced significant changes in the mRNA and protein expression of dopamine D1 and D2 receptors in the striatum, the structure which is most engaged in morphine withdrawal. Here, the expressions of dopamine D1 and D2 receptors were significantly higher in comparison with morphine-withdrawal rats, but lower in comparison with fluoride-treated rats. In other structures, the expressions of dopamine D1 and D2 receptors were similar to those which were observed in morphine-withdrawal rats.

So far, the precise associations between fluoride toxicity and dopaminergic receptors are not recognized; however, the existing evidence on neurotoxic mechanisms of both opioids and fluoride suggests that synergistic effects are possible. For example, it has been documented that a long-term exposure to fluoride alters the neuronal morphology—it induces axon deterioration, myelin sheath degeneration and dark cells with scanty cytoplasm, and vacuolated swollen mitochondria in the neocortex, hippocampus, and cerebellum [31]. It results in the damage of cell organelles, such as mitochondria, and may alter the enzyme activity—alteration of calcium homeostasis, elevated levels of the Ca2+/CaM-dependent protein and kinase II gamma (CaMKIIg), increases in extracellular signal-regulated kinases (ERK), decreased expression of the anti-apoptotic protein Bcl-2, and increased caspase-3 activation [32,33]. Fluoride may enhance superoxide formation, inducing oxidative stress and lipid peroxidation [6,21], leading to apoptosis [34,35,36]. Finally, fluoride exposure may influence the neurotransmitter level. Fluoride increased the levels of epinephrine, histamine, serotonin, and glutamate and decreased levels of norepinephrine, acetylcholine, and dopamine in a dose-dependent manner. Tsunoda et al., (2012) documented fluoride influence on dopamine and serotonin metabolites [37].

On the other hand, long-term exposure to morphine influences mainly dopaminergic neurotransmission. As was described in the Introduction, morphine administration via opioid and GABAergic receptors increases the dopamine level while morphine discontinuation produces the opposite effect. It has been demonstrated that higher levels of dopamine (associated with chronic morphine exposure) may act as a neurotoxic agent [38]. Dopamine is easily oxidized and may induce oxidative stress in cells. The mitochondrial enzyme MAO-B induces deamination of dopamine, producing DOPAC and hydrogen peroxide (H_2_O_2_). Additionally, dopamine auto-oxidation produces the superoxide anion O_2_– and H2O2. Both O_2_– and H_2_O_2_ generate highly toxic radicals which disturb cellular processes. This hypothesis was confirmed by Guzmán et al. (2006) and Özmen et al. (2007), who showed that morphine decreased the level of reduced glutathione (GSH) [39,40].

Taking into account the above-mentioned data and the results of the present study, it can be concluded that pre- and postnatal exposure to fluoride may disturb the activity of brain and may directly influence the effect of long-term exposure to morphine. It seems that the striatum and the cerebellum were structures damaged the most in the present study.

## 4. Materials and Methods

The study was approved by the local Ethics Committee of the Medical University in Lublin (No. 20/2014, approval date 10 November 2014) in accordance with the Directive 2010/63/EU on the protection of animals used for scientific purposes.

### 4.1. Fluoride Toxicity Procedure

The study was conducted on Wistar rats. Optimum conditions for the rats were ensured, i.e., a natural day–night cycle and room temperature of 22 ± 1 °C. Mature females were mated with males and then isolated and divided into two groups: the control and experimental. Females from the experimental group (*n* = 3) received 50 ppm sodium fluoride solution (NaF) in drinking water ad libitum, starting from the first day of gestation. Fluoride dose was adjusted to represent its concentration in rat’s blood similar to that observed in the serum of people environmentally exposed to fluoride compounds [13,14]. The solution of NaF was prepared daily. Pregnant females from the control group (*n* = 3) received tap water until offspring were weaned. The offspring were fed by the mothers, and those from the experimental group received NaF in drinking water ad libitum. On the 21st post-natal day (PND 21) the offspring were placed in separate cages. From that moment, the young rats still received NaF until PND 60. The control animals received tap water ad libitum in the same period.

### 4.2. The Procedure of Morphine Dependence

The state of morphine dependence was induced in adult rats starting from the 60th PND, both in fluoride-treated rats and the saline-treated (control group) animals. This was obtained by administering increasing doses of morphine (10.0, 15.0, 20.0, 25.0, 30.0, 35.0, 40.0, 50.0 mg/kg, ip) twice a day, for eight consecutive days (PND 60–67). On the next day (PND 68), the next dose of morphine (50.0 mg/kg) was administered, and 1 h later, an opioid receptor antagonist—naloxone (2.0 mg/kg, ip)—was injected in order to induce morphine withdrawal. Next, the animals were immediately placed in 10-litre glass cylinders. The number of jumps was measured for a period of 30 min. Immediately after the behavioural experiments, the rats were decapitated, and the brains were quickly removed. Four brain structures (striatum, hippocampus, prefrontal cortex, and cerebellum) were selected for neurochemical and molecular experiments and microscope observation. The obtained tissue was immediately frozen using liquid nitrogen. The samples were stored at −80 °C for neurochemical analysis and Western blot analysis. The samples for the quantitative real-time PCR reaction were placed in Trizol reagent (Invitrogen, Carlsbad, CA, USA) and stored at −80 °C. The samples of brain tissues for the IHC reaction were fixed in formalin.

In the study, the animals were divided into the following groups:(1)*saline group*—rats receiving 0.9% NaCl (*n* = 6);(2)*saline + naloxone group*—rats receiving 0.9% NaCl from the prenatal period until adulthood; on the last day of the experiment, they received saline with naloxone (*n* = 6); these animals were excluded from molecular studies;(3)*fluoride group*—rats treated with fluoride (NaF) from the prenatal period until adulthood and receiving 0.9% NaCl instead of morphine. They received naloxone on the last day of the experiment (*n* = 6);(4)*morphine group*—rats receiving 0.9% NaCl from the prenatal period until adulthood, and then procedure of morphine dependence was performed (*n* = 6);(5)*fluoride + morphine group*—rats treated with fluoride (NaF) from the prenatal period until adulthood, F-treated, and then the procedure of morphine dependence was performed (*n* = 6).

### 4.3. Ex Vivo Biochemical Studies—the Analysis of the Concentration of Dopamine and Its Metabolites in the Striatum, the Hippocampus, the Prefrontal Cortex, and the Cerebellum by the HPLC-ED Method 

The concentrations of dopamine and its metabolites were measured using high-pressure liquid chromatography with electrochemical detection. The mobile phase was 0.05 M citrate-phosphate buffer of pH 3.5, 0.1 mM of EDTA, 1 mM of sodium octyl sulfonate, and 3.5% methanol. The flow rate was 1 ml/min. Prior to homogenization with 0.1 M perchloric acid containing 0.05 mM of ascorbic acid, all samples were accurately weighted. Then, the samples were centrifuged and filtered using a cellulose membrane (RC 58 0.2-im). The assay was conducted with the use of an HP 1050 (Hewlett-Packard, Golden, CO, USA) equipped with C18 columns. For the purpose of determining the concentration of dopamine and its metabolites, the analysis of standards was conducted in parallel with the analysis of the samples, and then followed by comparison of the obtained peaks.

### 4.4. The Analysis of Gene Expression in the Striatum, the Hippocampus, the Prefrontal Cortex, and the Cerebellum by Real-Time Quantitative Reverse Transcription PCR (RQ-PCR)

Quantitative mRNA expression of D1R (dopamine receptor D1) and D2R (dopamine receptor D2) genes was performed in a two-step reverse transcription PCR. The GAPDH gene was used as a reference gene. RNA was isolated from the tissue samples kept at −80 °C using the commercial kit RNeasy MiniKit (Qiagen, Hilden, Germany) according to the manufacturer’s instructions. The quality and quantity of RNA were determined with a NanoDrop ND 1000 (Thermo Fisher Scientific™, Waltham, MA, USA). The obtained matrix was transcribed into cDNA with the Omniscript RT Kit (Qiagen, Hilden, Germany) according to the manufacturer’s instructions. Quantitative real-time CPR was conducted using a 7500 Fast Real-Time PCR System (Applied Biosystems, Foster City, CA, USA) and the reagent Power SYBR Green PCR Master Mix (Applied Biosystems, Foster City, CA, USA), i.e., the buffer solution containing AmpliTaq Gold DNA polymerase, mixture of deoxynucleotides, SYBR Green dye, and ROX reference dye (normalisation of the fluorescent signal). Monitoring the real-time increase in the CPR reaction product was made possible by fluorescence measurement which is proportional to the concentration of the product in a mixture. For the purpose of further calculations, the mean from the two measurements was used. To calculate the values, the ΔΔCt relative quantification method was used. From the means of the logarithmic expression values, fold change between groups was calculated.

The following primer pairs were used: GAPDH forward: ATG ACT CTA CCC ACG GCA AG, reverse: CTG GAA GAT GGT GAT GGG TT; D1R forward: CGC GTA GAC TCT GAG ATT CTG AAT T, reverse: GAG TTA AGG AGC CAC CAC ATC AGT; D2R forward: TGA CAG TCC TGC CAA ACC AGA GAA, reverse: A TGG GCA TGG TCT GGA TCT CAA AGA.

### 4.5. The Analysis of Protein Expression in the Striatum, the Hippocampus, the Prefrontal Cortex, and the Cerebellum by the Western Blotting Method

Tissue sample homogenates were treated with RIPA lysis buffer containing protease and phosphatase inhibitors (cOmplete™, Mini Protease Inhibitor Cocktail, Roche, Switzerland, PhosSTOP™, Roche, Switzerland). Protein concentration in the filtrate was determined with the BCA method using a commercial Pierce™ BCA Protein Assay Kit (Thermo Fisher Scientific™, Waltham, MA, USA). Then, electrophoretic protein fractionation was conducted (SDS-PAGE) using 10% polyacrylamide gel by placing 30 μg protein/well. The fractionated proteins were transferred onto a 0.2 μm PVDF membrane (Thermo Fisher Scientific™, Waltham, MA, USA) by a wet transfer. Prior to incubation with antibodies, the membranes were placed in a blocking buffer (5% BSA) for 60 min. The brain protein expression was detected using an antibody against D1R (ab81296) and D2R (ab85367)—rabbit pAb (Abcam, Cambridge, UK) diluted 1:800 and 1:1000, respectively, and sAb goat anti-rabbit IgG HRP H&L (ab97051) (Abcam, Cambridge, UK). Expression of β-actin was detected using the β-Actin antibody: sc-47778 (Santa Cruz Biotechnology, US). The membranes were developed with an ECL Advance Western Blotting Detection Kit (GE Healthcare, Chicago, IL, USA), and subsequently, bands were visualized using the Molecular Imager ChemiDock XRS+ (Bio-Rad, Hercules, CA, USA).

### 4.6. Immunohistochemical Analysis of D1 and D2 Receptors in the Hippocampus, the Striatum, the Prefrontal Cortex, and the Cerebellum

The dissected whole brains (*n* = 6 in each studied group) were placed in 4% buffered formalin solution for 24 h. Subsequently, they were washed several times with distilled water and subjected to a dehydration series. The dehydrated samples were placed in paraffin for 3 h and locked in cases. The obtained blocks were cut using a microtome (HM 340E Electronic Rotary Microtome, Thermo Fisher Scientific™, Waltham, MA, USA) into sections of 3–5 µm, which were then placed on histological slides (3-aminopropyl-trietoxy-silane, cat.no. J2800AMNZ, Thermo Fisher Scientific™, Waltham, MA, USA). Subsequently, the sections were deparaffinised and hydrated. In order to expose epitopes, the deparaffinised sections were boiled twice in a microwave oven (700 W, 4 and 3 min.) in 10 nM citrate buffer (pH 6.0). Once cooled and washed with PBS, the preparations were incubated for 60 min at room temperature with a primary antibody against DR1 (cat. no.: sc-14001, SantaCruz Biotechnology, Dallas, TX, USA) and DR2 (cat. no.: sc-5303, SantaCruz Biotechnology, Dallas, TX, USA) in final solution (1:750). For the purpose of visualisation of the binding site of the primary antibodies, a commercial kit Dako LSAB + System-HRP (Dako Inc., Canpinteria, CA, USA) was used. The kit is based on the reaction with a biotinylated antibody (against rabbit and mouse proteins), streptavidin marked with peroxidase, and 3 3’ diaminobenzidine (DAB) as a chromogen. In this reaction, brown-coloured precipitate appears in the presence of the antigen. The kit was used according to the manufacturer’s instructions. The preparations were additionally stained with hematoxylin. To visualise the samples and obtain pictures, a light microscope (Leica DM5000 B, Wetzlar, Germany) integrated with a camera was used. For a negative control, specimens were processed in the absence of primary antibodies.

### 4.7. Statistical Analysis

The results of behavioural analyses were evaluated by analysis of variance (one-way ANOVA) using the Graph-Pad Prism Software 5.04. Post-hoc analysis was conducted with a Tukey test. The results are presented as mean values ± S.E.M.

The results of neurochemical and molecular analyses were statistically analysed with Statistica 13 software and are presented as mean values ± SD. The Shapiro-Wilk W test did not show conformity with the normal distribution; therefore, the non-parametric Mann–Whitney U test was used for comparison of the groups. Statistical significance was set at *p* < 0.05.

## 5. Conclusions

In the present study, it was shown for the first time that pre- and postnatal exposure to fluoride influenced the phenomenon of morphine dependence. Moreover, the dopaminergic disturbances within the striatum and the cerebellum played a predominant role as both the alterations of dopamine metabolism and profound alterations in striatal D1 and D2 receptors were discovered in these structures. The present study provides new insight into the global problem showing direct associations between environmental factors and addictive disorders.

## Figures and Tables

**Figure 1 ijms-21-02361-f001:**
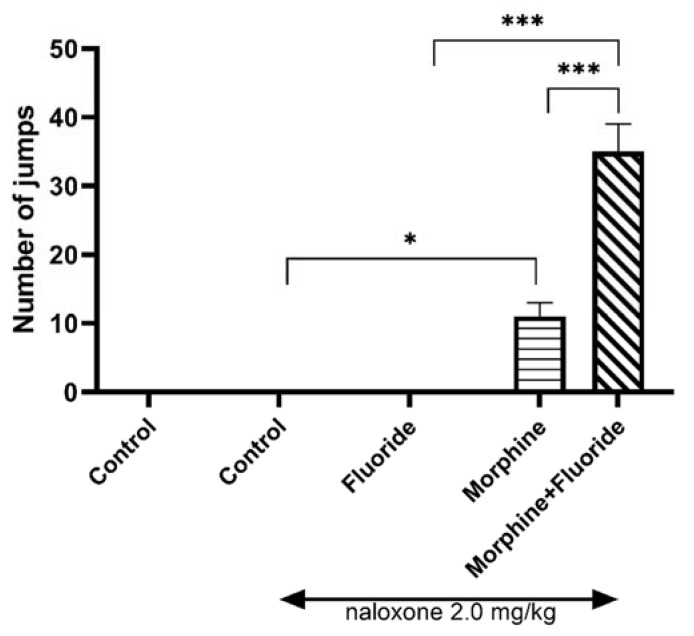
The number of jumps in morphine-dependent rats which were prenatally and postnatally exposed to fluoride. The animals were exposed to NaF solution from conception to the 60th postnatal day (PND). The morphine dependence was developed in these rats by administration of increasing doses of morphine for eight consecutive days. On the 68th PND, the intensity of naloxone-induced (2.0 mg/kg, ip) morphine withdrawal signs was assessed in studied rats. * *p* < 0.05, *** *p* < 0.001 (Tukey’s test).

**Figure 2 ijms-21-02361-f002:**
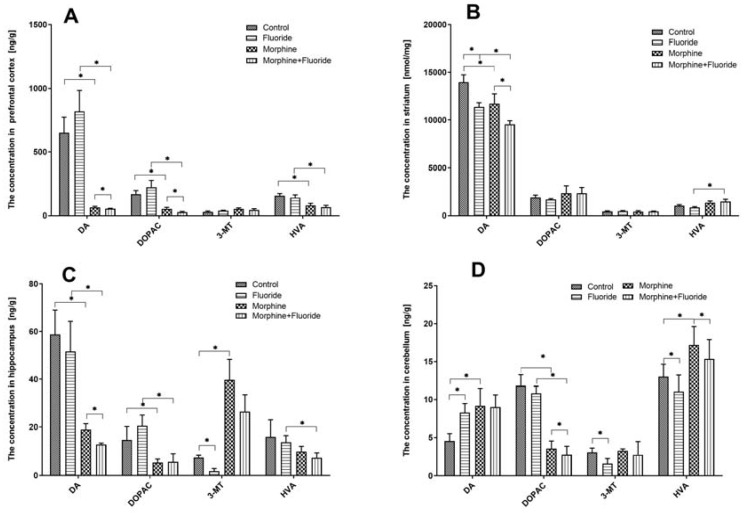
The concentrations of DA, DOPAC, HVA, and 3-MT (ng/g) in the prefrontal cortex (**A**), striatum (**B**), hippocampus (**C**), and cerebellum (**D**) of the rat brain in the control, fluoride, morphine, and morphine + fluoride groups. Rats were treated with 50 ppm of NaF and/or were administrated morphine in increasing doses (10–50 mg/kg). The results are presented as means ± SD. The statistical analysis was performed using the Mann-Whitney U test, * *p* < 0.05.

**Figure 3 ijms-21-02361-f003:**
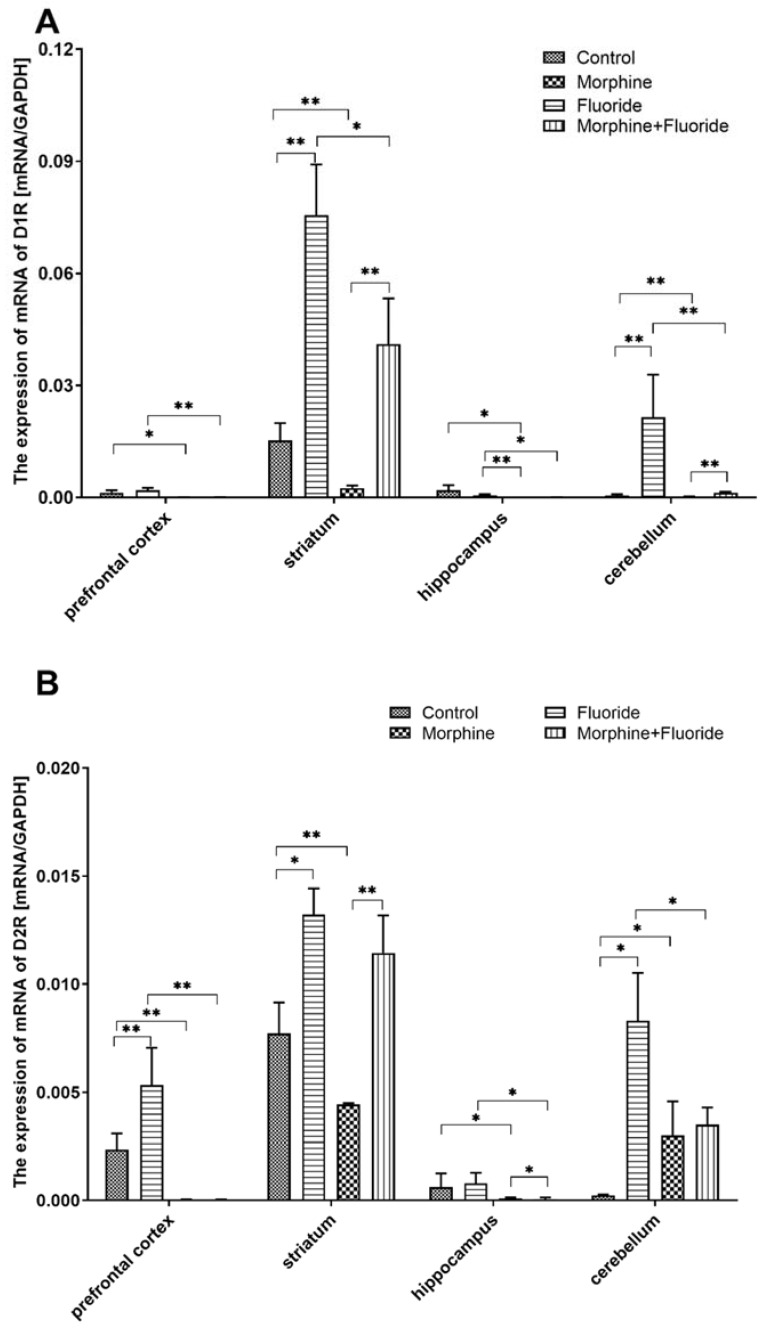
The expression of mRNA of D1R (**A**) and D2R (**B**) in the prefrontal cortex, striatum, hippocampus, and cerebellum of the rat brain in the control, fluoride, morphine, and morphine + fluoride groups. Rats were treated with 50 ppm of NaF and/or were administered morphine in increasing doses (10–50 mg/kg). The results are presented as means ± SD. The statistical analysis was performed using the Mann–Whitney U test * *p* < 0.05, ** *p* < 0.01.

**Figure 4 ijms-21-02361-f004:**
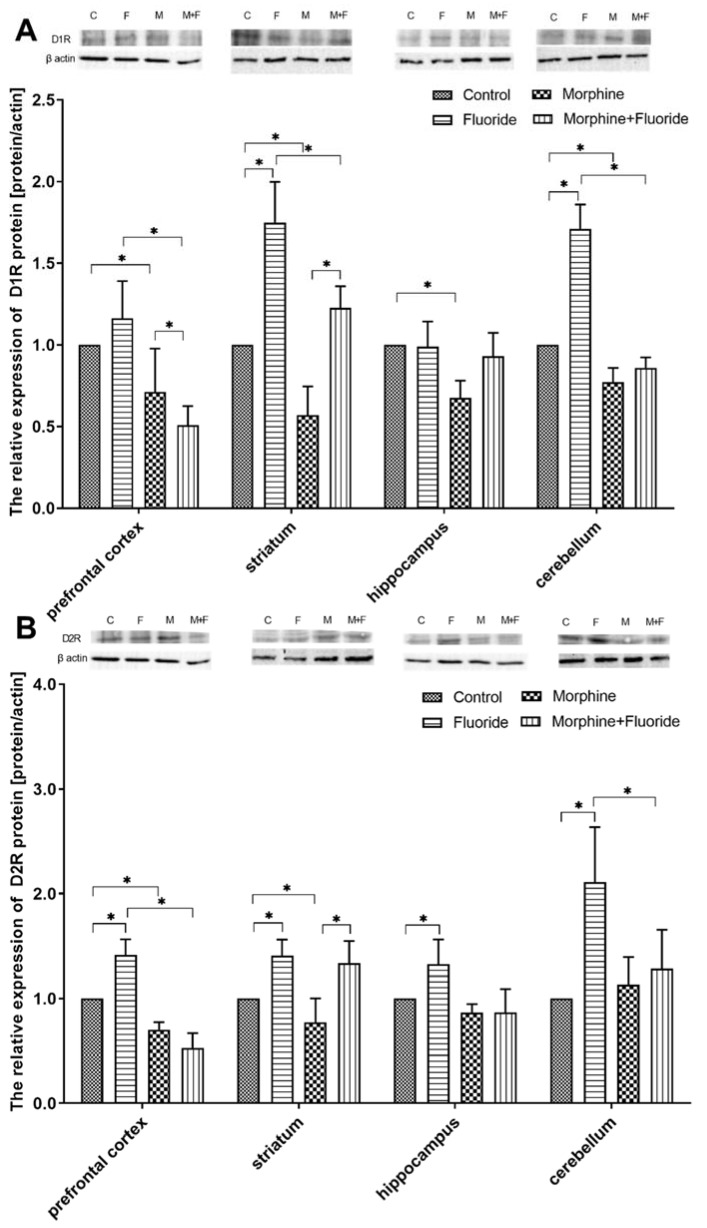
Representative Western blots and densitometric analysis of D1R (**A**) and D2R (**B**) protein expression levels (normalized to β-actin) in the brain of control (C), perinatally F-exposed (F), morphine administered (M), perinatally F-exposed, and morphine administered rats (F + M) in a morphine dependence experimental model in chosen brain structures. The results are expressed as means ± SD. * *p* < 0.05 (Mann–Whitney U test).

**Figure 5 ijms-21-02361-f005:**
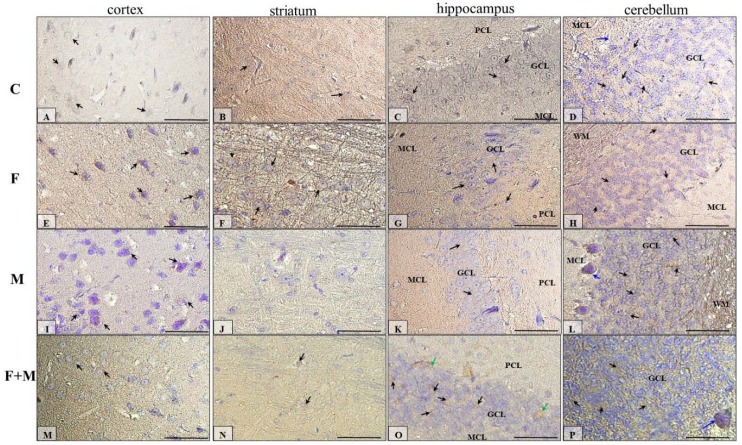
Representative microphotography showing immunoexpression of dopamine receptor D1 in the cortex (**A**,**E**,**I**,**M**), striatum (**B**,**F**,**J**,**N**), hippocampus (**C**,**G**,**K**,**O**), and cerebellum (**D**,**H**,**L**,**P**) of rats in the control (**A**–**D**), fluoride (**E**–**H**), morphine (**I**–**L**), and morphine and fluoride (**M**–**P**) groups. IHC reaction. Scale bar 50 µm. GCL—granular cell layer, MCL—molecular cell layer, PCL—polymorphic cell layer, WM—white matter. Black arrows—neurons of neocortex or striatum or GCL from the hippocampus or cerebellum, green arrows—neurons of PCL of the dentate gyrus, blue arrows—Purkinje cells of the ganglion cell layer of the cerebellar cortex.

**Figure 6 ijms-21-02361-f006:**
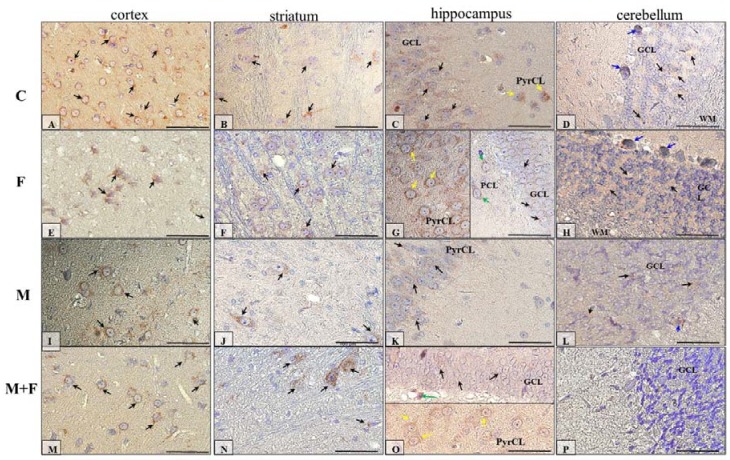
Representative microphotography showing immunoexpression of dopamine receptor D2 (D2R) in the cortex (**A**,**E**,**I**,**M**), striatum (**B**,**F**,**J**,**N**), hippocampus (**C**,**G**,**K**,**O**), and cerebellum (**D**,**H**,**L**,**P**) in rats from the control (**A**–**D**), fluoride (**E**–**H**), morphine (**I**–**L**), and morphine and fluoride (**M**–**P**) groups. IHC reaction. Scale bar 50 µm. GCL—granular cell layer, MCL—molecular cell layer, PCL—polymorphic cell layer, PyrCL—pyramidal cell layer, WM—white matter. Black arrows—neurons of the neocortex or striatum or GCL from the hippocampus or cerebellum, green arrows—neurons of the PCL of the dentate gyrus, yellow arrows—neurons of the PyrCL cornu ammonis, blue arrows—Purkinje cells of the ganglion cell layer of the cerebellar cortex.

**Table 1 ijms-21-02361-t001:** Summary of the expression of dopamine receptors (D1 and D2) in the control and study groups presented as intensity of immunostaining.

	Prefrontal Cortex		Striatum		Hippocampus		Cerebellum	
	D1	D2	D1	D2	D1	D2	D1	D2
Control	++	+	+	++	+	+	+	+
Fluoride	+++	+	++	+++	+	++	++	+++
Morphine	+	+	−/+	+	+	+	++	++
Morphine + Fluoride	+	+	+	++	+	+	+	+

Intensity of immunostaining scored as negative (−), weakly positive (+), moderately positive (++) or strongly positive (+++).

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
