# Peer review of "Fluoride Affects Dopamine Metabolism and Causes Changes in the Expression of Dopamine Receptors (D1R and D2R) in Chosen Brain Structures of Morphine-Dependent Rats"

_ijms, 2020, doi:10.3390/ijms21072361_

Round 1

Reviewer 1 Report

   Authors studies the effect of pre- and postnatal exposure to 50 ppm Fluoride on the initiation/development of morphine dependence. using animal model. They showed their evidence that fluoride exposure worsens the behavior disturbance of morphine withdrawal. They also showed that the dopamine metabolism as well as D1 and D2 receptors were disturbed in striatum and cerebellum. Perinatal fluoride exposure disturbed dopamine pathway and resulted in worse morphine withdrawal syndrome .  The findings of this study is interesting and may influence the policy of adding fluoride to tap water.

There are several issues need to be clarified:

Several representative pictures of western blot (figure 4-A) have darker background as compared with the others. Unequal quality of western blot may decrease the confidence level of quantitative analysis of western blot. Authors should provide better representative pictures to convince readers.

There was no quantitative analysis for the immunohistochemistry studies presented in this article. Quantitative data should be provided to convince the readers about the results authors claimed.

All the experimental groups for morphine dependence received Naloxone to induced withdrawal syndrome, so it was improper to draw the conclusions that the changes in dopamine metabolism and receptors are induced by morphine withdraw. Therefore, several conclusions claimed in discussion section: Line 316 to 317; Line 350 to 352; Line 359 to 361 should be modified

Author Response

Comments and Suggestions for Authors

   Authors studies the effect of pre- and postnatal exposure to 50 ppm Fluoride on the initiation/development of morphine dependence. using animal model. They showed their evidence that fluoride exposure worsens the behavior disturbance of morphine withdrawal. They also showed that the dopamine metabolism as well as D1 and D2 receptors were disturbed in striatum and cerebellum. Perinatal fluoride exposure disturbed dopamine pathway and resulted in worse morphine withdrawal syndrome.  The findings of this study is interesting and may influence the policy of adding fluoride to tap water.

There are several issues need to be clarified:

Several representative pictures of western blot (figure 4-A) have darker background as compared with the others. Unequal quality of western blot may decrease the confidence level of quantitative analysis of western blot. Authors should provide better representative pictures to convince readers.

Thank you very much for this comment. Of course, we took into account the background differences, however the software cuts off the background (ImageLab, BioRad) and we do not compare the density of bands between structures.

There was no quantitative analysis for the immunohistochemistry studies presented in this article. Quantitative data should be provided to convince the readers about the results authors claimed.

According to Reviewer remark , the changes in intensity of IHC reaction now is presented in additional Table as -; +/-; +; ++;+++. Using the following scale, slides were evaluated by two independent histologists.

All the experimental groups for morphine dependence received Naloxone to induced withdrawal syndrome, so it was improper to draw the conclusions that the changes in dopamine metabolism and receptors are induced by morphine withdraw. Therefore, several conclusions claimed in discussion section: Line 316 to 317; Line 350 to 352; Line 359 to 361 should be modified –

According to Reviewer remark we corrected the sentences and we added “naloxone-induced morphine withdrawal…”

Reviewer 2 Report

Results: Suggest to make a list of the specific points that the authors want to convey and refer them to the figures.

IHC results Fig 5 and 6. The microphotographs are of limited resolution and the interpretation is beyond to what can actually be seen on the plates presented, even for a trained eye. The composites of photos are too busy, too many photos in the same plate, too many arrows showing too many findings. Suggest to rearrange and focus on the main findings placing the photos to be compared side by side. Please improve resolution and for very detail showing can insert a photo taken at a higher magnification. These were taken at 40X.

Please answer the following questions regarding the methods used for IHC which are essential for the interpretation of ICH studies

Slides were stained and processed simultaneously with the same batch of Ab at the same dilution?

Were the samples for IHC taken simultaneously than those for biochemical tests?

Were positive and negative controls carried out for the staining procedure itself?. E.g. slides omitting the 1st antibody, absorption test for the 1st antibody. Inclusion of a slide with known positive staining.

How many tissue blocks were analyzed for each region and how many sections were prepared, stained and examined.

Quantitative and qualitative methods to assess staining intensity and distribution?

Although it can be guessed please specified the counter stain used.

Line 245 fig 6E describe the specific changes observed and which additional staining method was used to assess basic histopathology. Here it seems to be artefacts induced during processing how the authors ruled out artefacts versus induced changes?. Apoptosis is described in association with fluoride toxicity? Any of the multiple stains used to assess apoptosis were used?.

In the discussion separate the histopathological changes described in the literature from those found in the study reported here. Lines 374-379 revised please. Specify in the results which are those changes and what stain was used to visualized them.

Methods. Fluoride toxicity procedures. Method used to adjust the dose? As indicated in line 409. Was fluoride measured in milk?. The fluoride was administered orally added to the water, drinking at libitum. How exposure was calculated?

Abstract and conclusions are very generic. There are specific findings to close up the statement e.g.

this study provides new evidences linking fluoride toxicity and morphine addiction using a rat model previously described, with pre- and post-natal oral exposure. It opens further avenues on the possible interplay between environmental toxins and chemical addictions.

Introduction leave it as such. Do not add at the end of the introduction what is shown in the results. Suggest to delete lines 85 and 86.

Author Response

Comments and Suggestions for Authors

Results: Suggest to make a list of the specific points that the authors want to convey and refer them to the figures.

IHC results Fig 5 and 6. The microphotographs are of limited resolution and the interpretation is beyond to what can actually be seen on the plates presented, even for a trained eye.

According to Reviewer remark we the IHC microphotography have been improved.

The composites of photos are too busy, too many photos in the same plate, too many arrows showing too many findings. Suggest to rearrange and focus on the main findings placing the photos to be compared side by side.

According to Reviewer remark we the number of arrows was reduced, however we prefer to leave the composition of photos.

Please improve resolution and for very detail showing can insert a photo taken at a higher magnification. These were taken at 40X.

According to Reviewer remark the resolution has been improved (now the each photo is 2 times bigger, and the scale bar is 50µm).

Please answer the following questions regarding the methods used for IHC which are essential for the interpretation of ICH studies

Slides were stained and processed simultaneously with the same batch of Ab at the same dilution?

Yes, tissue on the slides were covered by the antibodies in the same concentration.

Were the samples for IHC taken simultaneously than those for biochemical tests? ?

Yes, a half of brain was dropped into formalin, second half part of brain was frozen in liquid nitrogen

Were positive and negative controls carried out for the staining procedure itself?. E.g. slides omitting the 1st antibody, absorption test for the 1st antibody. Inclusion of a slide with known positive staining.

Yes, there were made negative control (omitting the 1st antibody); brains from control animals were recognized as a positive control because dopamine receptors are expressed in normal nervous tissue therefore, we considered unnecessary inclusion of photos with known positive staining.

How many tissue blocks were analyzed for each region and how many sections were prepared, stained and examined.

Six of each group (saline group, saline+naloxone group, fluoride group, morphine group ) paraffin brain sections were stained and then evaluated.

Quantitative and qualitative methods to assess staining intensity and distribution?

Now the changes in intensity of IHC reaction now is presented additional Table as -; +/-; +; ++;+++. Using the following scale, slides were evaluated by two independent histologists.

Although it can be guessed please specified the counter stain used.

According to Reviewer remark we improved within the text body of manuscript.

Line 245 fig 6E describe the specific changes observed and which additional staining method was used to assess basic histopathology. Here it seems to be artefacts induced during processing how the authors ruled out artefacts versus induced changes?. Apoptosis is described in association with fluoride toxicity? Any of the multiple stains used to assess apoptosis were used?.

Unfortunately, no staining was performed to show apoptosis, therefore the sentence describing this phenomenon was removed from the text.

In the discussion separate the histopathological changes described in the literature from those found in the study reported here. Lines 374-379 revised please.

Specify in the results which are those changes and what stain was used to visualized them.

According to Reviewer remark we ccorrected the sentence

Methods. Fluoride toxicity procedures. Method used to adjust the dose? As indicated in line 409. Was fluoride measured in milk?. The fluoride was administered orally added to the water, drinking at libitum. How exposure was calculated?

The quantity of drunk NaF treated water per cage was measured every day The fluoride concentration was not measured in milk but in previous studies using the same model of fluoride intoxication the level of F was estimated in brain structures and serum (Dec, K et al. 2019; and data in publish).

Abstract and conclusions are very generic. There are specific findings to close up the statement e.g.this study provides new evidences linking fluoride toxicity and morphine addiction using a rat model previously described, with pre- and post-natal oral exposure. It opens further avenues on the possible interplay between environmental toxins and chemical addictions.

According to Reviewer remark we corrected the sentence

Introduction leave it as such. Do not add at the end of the introduction what is shown in the results. Suggest to delete lines 85 and 86.

According to Reviewer remark we corrected the sentence

Round 2

Reviewer 2 Report

Great improvement. Good work.

Author Response

The authors frequently do not make difference in nomenclature between
fluorine element (it is a very toxic gas) and fluoride ion! At least in the
Abstract (line 30, 34 and 37) should be written "fluoride ion" replacing
"fluoride". - We corrected the sentence according to Reviewer remark

A big mistake is in line 47: the sentence "Fluoride is a neurotoxic element"
is wrong! The element calls fluorine and it is a deadly toxic gas. Fluoride
ion is toxic, but it is not an element, it is an ion. The sentence in line 47
after correction could be:
  "Fluoride is a neurotoxic ion."..etc. or "Fluoride ion is
neurotoxic"....etc - We corrected the sentence according to Reviewer remark

Use the nomenclature "fluorine" and "fluoride" correctly: fluorine is the
name of an  element (a toxic gas) and fluoride is an anion with a negative
charge

We are very grateful for this remark.